# Preclinical Assessment of ADAM9-Responsive Mesoporous Silica Nanoparticles for the Treatment of Pancreatic Cancer

**DOI:** 10.3390/ijms241310704

**Published:** 2023-06-27

**Authors:** Etienne J. Slapak, Mouad el Mandili, Marieke S. Ten Brink, Alexander Kros, Maarten F. Bijlsma, C. Arnold Spek

**Affiliations:** 1Laboratory of Experimental Oncology and Radiobiology, Center for Experimental and Molecular Medicine, Amsterdam UMC Location University of Amsterdam, Meibergdreef 9, 1105 AZ Amsterdam, The Netherlands; e.j.slapak@amsterdamumc.nl (E.J.S.); m.elmandili@amsterdamumc.nl (M.e.M.); m.s.tenbrink@amsterdamumc.nl (M.S.T.B.); m.f.bijlsma@amsterdamumc.nl (M.F.B.); 2Oncode Institute, 3521 AZ Amsterdam, The Netherlands; 3Cancer Center Amsterdam, Cancer Biology, 1081 HV Amsterdam, The Netherlands; 4Department of Supramolecular & Biomaterials Chemistry, Leiden Institute of Chemistry, Leiden University, 2333 CC Leiden, The Netherlands; a.kros@chem.leidenuniv.nl

**Keywords:** MSN, PDAC, targeted therapy, drug delivery, neurotoxicity, leukopenia, antitumor

## Abstract

Pancreatic adenocarcinoma (PDAC) remains largely refractory to chemotherapeutic treatment regimens and, consequently, has the worst survival rate of all cancers. The low efficacy of current treatments results largely from toxicity-dependent dose limitations and premature cessation of therapy. Recently, targeted delivery approaches that may reduce off-target toxicities have been developed. In this paper, we present a preclinical evaluation of a PDAC-specific drug delivery system based on mesoporous silica nanoparticles (MSNs) functionalized with a protease linker that is specifically cleaved by PDAC cells. Our previous work demonstrated that ADAM9 is a PDAC-enriched protease and that paclitaxel-loaded ADAM9-responsive MSNs effectively kill PDAC cells in vitro. Here, we show that paclitaxel-loaded ADAM9-MSNs result in off-target cytotoxicity in clinically relevant models, which spurred the development of optimized ADAM9-responsive MSNs (OPT-MSNs). We found that these OPT-MSNs still efficiently kill PDAC cells but, as opposed to free paclitaxel, do not induce death in neuronal or bone marrow cells. In line with these in vitro data, paclitaxel-loaded OPT-MSNs showed reduced organ damage and leukopenia in a preclinical PDAC xenograft model. However, no antitumor response was observed upon OPT-MSN administration in vivo. The poor in vivo antitumor activity of OPT-MSNs despite efficient antitumor effects in vitro highlights that although MSN-based tumor-targeting strategies may hold therapeutic potential, clinical translation does not seem as straightforward as anticipated.

## 1. Introduction

Pancreatic cancer (PDAC) has a 5-year overall survival of less than 9% [1,2], making PDAC the deadliest of all common cancers. To a large extent, the poor outcome can be attributed to the lack of effective (chemotherapeutic) treatments. Although significant progress has been made in the treatment of cancer in general, improvements in PDAC have lagged, and resection remains the only treatment with curative potential. Current first-line treatment options in non-resectable PDAC include systemically administered gemcitabine with nab-paclitaxel and FOLFIRINOX; however, both options are characterized by toxicity-dependent dose limitations, resulting in treatment discontinuation in up to 60–70% of patients [3]. Common side effects include nausea, fatigue, diarrhea, neutropenia, and neuropathy.

Nanocarriers have shown promise in reducing systemic toxicity by offering targeted delivery of chemotherapeutics to cancer cells without affecting healthy non-tumor cells. Among the most frequently reported drug nanocarriers in PDAC are mesoporous silica nanoparticles (MSNs) [4]. MSNs are characterized by their low toxicity profile in vivo, a high loading capacity due to their porous structure, high biocompatibility, and potential for separate inner-core and outer surface modifications [5,6]. One particularly interesting feature of MSNs is the ability to modify the carrier with a *gatekeeper* system, enabling the spatiotemporally controlled release of (drug) cargo and potentially reducing side effects by specifically targeting tumor cells.

In PDAC, several different gatekeeper systems have been reported [4], of which protease-activity-dependent gatekeepers are especially interesting due to the abundant stroma that features high expression of numerous proteases [7,8,9]. Protease-activity-dependent gatekeepers consists, in short, of a peptide linker responsive to tumor-enriched proteases coupled with biotin that is attached to the outer surface of the MSN, which, after drug loading, is sealed by the interaction of biotin with avidin. Theoretically, upon reaching the tumor, the protease-responsive peptide linker is cleaved, and the loaded drugs are released. Previously reported protease activity-dependent gatekeepers were developed around MMP2 and MMP9 activity, as these proteases are often overexpressed in cancer [7,10]. To identify a truly PDAC-specific enriched protease, we previously performed an unbiased in silico analysis and identified the membrane-anchored member of the zinc protease superfamily ADAM9 (a disintegrin and metalloprotease domain 9) as a likely PDAC-specific candidate [9]. Next, we showed that ADAM9, which participates in a variety of physiological functions through the disintegrin domain for adhesion and the metalloprotease domain for ectodomain shedding of a wide variety of cell surface proteins [11], efficiently cleaved linker peptides containing the putative ADAM9 cleavage site SPLAQAVRSSK, based upon which we generated MSNs containing the biotin-coupled ADAM9-peptide linker. These ADAM9-responsive MSNs (ADAM9-MSNs) loaded with paclitaxel were ultimately shown to be effective against PDAC cells without affecting monocyte cell viability [9]. 

In the current manuscript, which aimed to assess the preclinical potential of our ADAM9-MSNs. However, we show that paclitaxel-loaded ADAM9-MSNs result in off-target cytotoxicity in clinically relevant models associated with paclitaxel administration. Therefore, we developed an optimized ADAM9-responsive MSN (OPT-MSN) and assessed the potential clinical applicability of these OPT-MSNs by addressing the potential cytotoxicity on key cell types involved in neurotoxicity and leukopenia, as well as their efficacy in a PDAC xenograft model.

## 2. Results

### 2.1. ADAM9-MSNs Do Not Reduce Paclitaxel-Induced Side Effects

We previously identified ADAM9 as a PDAC-enriched protease and reported that ADAM9-responsive paclitaxel-loaded MSNs efficiently kill PANC-1 cells in vitro but not (leukemic) monocytes [9]. Despite showing that limited drug release after preincubation with monocytes is important, the clinical implications of novel formulations are best assessed by focusing on the commonly reported side effects caused by systemic paclitaxel, such as neuropathy and leukopenia. To assess these cytotoxic effects of the previously established ADAM9-MSNs, we employed SH-SY5Y cells, which are among the most commonly used cell lines to test neuronal function and differentiation [12]. Administration of paclitaxel-loaded ADAM9-MSNs efficiently killed PANC-1 cells (Figure 1B) but also resulted in substantial cell death in SH-SY5Y when treated with similar amounts of ADAM9-MSNs (Figure 1D). It is important to mention that although SY5Y cells are less sensitive to paclitaxel than PANC-1 (Figure 1C,A, respectively) dose-responsive cytotoxicity was observed.

In a similar way, colony-forming assays on bone-marrow-derived cells can be used to assess the myelotoxicity of chemotherapy regimens [13]. Bone marrow cells were also as efficiently killed at concentrations approaching the IC20 for PANC-1—comparable to a concentration of 20 nM free paclitaxel (Figure 1E). Together, these results indicate that our ADAM9-MSNs also release their cytotoxic contents in undesired cellular contexts, which suggests that the ADAM9-MSNs are most likely to not reduce neuropathy and leukopenia in a clinical setting.

### 2.2. MSNs with a Modified ADAM9 Linker Have Increased PDAC Specificity and Reduced General Toxicity

The lack of specificity of the ADAM9-MSNs led us to critically evaluate the ADAM9-responsive linker. Although our ADAM9 linker is a well-known substrate to test the cleavage activity of ADAM9, several related proteases, such as ADAM8, −10, and −17 and MMP1, −2, −3, −8, −9, −12, and −14, may also cleave the peptide [14]. If any of these proteases is not strictly specific to PDAC, one could envision that off-target cleavage and subsequent drug release may ensue. To address this, we compared expression levels of the relevant proteases in PDAC: neuronal and bone marrow cells. Interestingly, this analysis revealed that human blood-derived M1 and M2 macrophages, as well as SH-SY5Y cells, express higher levels of MMP1, −2, −8, −9, −12, and −14 compared to PDAC cell lines (Figure 2A). The expression patterns of ADAM8, −10, and −17 are relatively similar between PDAC cell lines, blood-derived M1 and M2 macrophages, and SH-SY5Y cells, and as expected, ADAM9 is expressed at lower levels in blood-derived M1 and M2 macrophages and SH-SY5Y cells (Figure 2A). The housekeeping gene beta-actin (ACTB) was plotted alongside to confirm that expression levels derived from several different datasets could be confidently compared. These analyses indicate that several proteases capable of cleaving the ADAM9 linker are relatively ubiquitously expressed, which may explain the observed cell death of SH-SY5Y and reduced outgrowth of bone marrow colonies in the presence of ADAM9-MSNs.

To reduce cytotoxicity and to increase PDAC specificity, we next generated MSNs containing a linker peptide including a recently described more specific ADAM9 cleavage site, henceforth referred to as OPT-MSN and OPT linker [15] (Figure 2B). The development of the OPT linker is based on two-dimensional substrate mapping experiments that revealed that substitution of valine and arginine with homophenylalanine and threonine, as the cleavage moiety increases specificity for ADAM proteins [16]. The OPT linker is processed at least 18 times more efficiently by ADAM9 compared to the previous ADAM9 linker. However, most importantly, the OPT linker is processed less efficiently by ADAM8 and MMP1, −2, −8, −9, −12, and −14 [15].

In addition, we developed an uncleavable OPT-MSN (UNCL-MSN) linker by substituting the L-isoforms of the amino acids homophenylalanine and threonine bordering the cleavage site with their corresponding D-isoforms. The mirrored isoforms render ADAM9 incapable of recognizing and cleaving the substrate, thereby functioning as a control of the specificity of the system in general. As shown in Figure 2C, the new MSNs were generated successfully, as evidence by ZP and FT-IR measurements (Table 1, Figure 2D). As a result of surface amine grafting, ZP increased from −18.93 ± 0.12 mV to 19.13 ± 0.73 mV, followed by a decrease to 0.56 ± 4.26 mV following modification with COOH-PEG_4_-N_3_. Functionalization with the biotin-conjugated OPT-ADAM9 and UNCL linkers resulted in an increase in ZP to 30.53 ± 1.24 mV and 29.27 ± 1.096 mV, respectively. FT-IR spectra (Figure 2D) show clear peaks at 800 and 1055 cm^−1^, confirming Si-O-Si bonds, and absorbance at 950 cm^−1^ and 3280 cm^−1^, confirming silanol groups. Surface amine grafting results in a decrease in silanol absorbance at 950 cm^−1^ and 3280 cm^−1^, confirming the successful amination of the MSNs. PEG_4_-N_3_ surface modification resulted in FT-IR absorbance at 1640 cm^−1^ (carbonyl stretch), confirming secondary amide bond formation. Furthermore, peptide linker functionalization results in an increase in absorbance in the amine (1530 cm^−1^) and carbonyl region (1640 cm^−1^) coming from the peptide backbone and side chains. A complete overview of the DLS, FT-IR, and ZP measurements of all intermediate products is presented in Table 1.

To confirm successful loading and protease-dependent drug release, we first assessed the cytotoxic potential of OPT-MSN and UNCL-MSNs on PANC-1 and Capan-2 PDAC cells. After 72 h, efficient dose-dependent cell killing was observed in both cell lines exposed to OPT-MSNs (Figure 2E,F). UNCL-MSN-treated PANC-1 cells did not show significant cell death (Figure 2E). At 1.5 µg of PTX-loaded OPT-MSNs, 80% of PANC-1 cells were killed, whereas PANC-1 cells treated with a similar amount of UNCL-MSNs showed no signs of cell death. These findings confirm that paclitaxel-loaded MSNs are tightly capped and that no leakage of loaded paclitaxel occurs. More importantly, the results also show that paclitaxel release specifically occurs via the proteolytic activity of ADAM9. As shown in Figure 1D,E, the treatment of bone marrow and SH-SY5Y cells with the previously developed, unoptimized ADAM9-MSNs resulted in significant non-cancer-specific cell death. We repeated those experiments with OPT-MSNs and observed no neurotoxicity or myelosuppression in vitro. At concentrations killing 80% of PANC-1 cells, no cell death was observed in SH-SY5Y cells (Figure 2G). Similarly, the number of colonies of untreated bone marrow cells was almost equal to that of colonies treated with an equivalent of OPT-MSNs that killed 80% of PANC-1 cells (Figure 2H). These findings imply that our OPT-MSN might reduce clinically relevant adverse effects caused by paclitaxel while efficiently killing PDAC cells. 

### 2.3. Biodistribution of MSNs In Vivo

After showing highly specific PDAC cell death by paclitaxel-loaded OPT-ADAM9-MSNs in vitro, we moved on to test their in vivo efficacy. To this end, we first assessed the biodistribution of intravenously injected fluorescently labeled MSNs. We developed ATTO488-labeled MSNs (ATTO-MSN) that can be detected by FACS. Next, we generated PDAC tumors by subcutaneously injecting 1 × 10^6^ KP2 cells in mice, and upon reaching a size of 300 mm^3^, 12.5, 25, or 50 mg/kg, ATTO-MSNs were injected in the tail vein. Four hours later, the mice were killed, and the tumors were analyzed. A dose-dependent number of MSNs was detected in the tumors, with the highest dose yielding a fourfold increase compared to the lowest dose (Figure 3A). Using the highest dose, we next aimed to determine the optimal dosing schedule resulting in the highest accumulation of MSNs in the tumor. After generating subcutaneous tumors, three groups of mice received a tail injection of 50 mg/kg ATTO-MSNs and were killed after either 24, 48, or 72 h, with the 72 h group receiving a second dose of 50 mg/kg ATTO-MSNs after 48 h. The increase in the circulation time of the ATTO-MSNs from 4 h to 24 h resulted in an increase in ATTO-MSNs detected at the tumor site (Figure 3B). Another 24 h later (t = 48 h), the abundance of ATTO-MSNs slightly decreased, followed by an increase after the second injection (t = 72 h), indicating that a second injection substantially increased the number of MSNs in the tumor (Figure 3B). Together, these results show that repeated injection of 50 mg/kg MSNs resulted in high intratumoral MSN levels.

### 2.4. Antitumor Efficacy of OPT-MSNs In Vivo

Having established the dosing regimen that achieves high MSN numbers in subcutaneous tumors, we finally assessed the efficacy of paclitaxel-loaded OPT-MSNs in the same tumor model. First, we ascertained that KP2 cells were capable of unlocking the OPT-MSNs and that the released paclitaxel killed KP2 cells in vitro. Although less sensitive than PANC-1 cells, a dose-dependent response was observed (Figure 4A)—a trend similar to that of free paclitaxel administration (Figure 1A and Appendix A). The in vivo antitumor efficacy and reduction in side effects were subsequently assessed by treating the mice with a total of seven injections of 50 mg/kg paclitaxel-loaded OPT-MSNs over the course of 3 weeks, whereas control mice received either PBS or 20 mg/kg free paclitaxel (dose based on [17]). The weight and tumor volume of the mice were closely monitored during the experiment. Two out of three mice treated with free paclitaxel showed severe side effects, resulting in a decrease in weight (Figure 4B) that required culling of the mice before the end of the experiment. Over the course of the experiment, no reduction in tumor growth was observed in the OPT-MSN-treated group compared to the control group (Figure 4C). Interestingly, treatment with free PTX originally caused a delay in tumor growth; however, once the tumor started growing exponentially, paclitaxel treatment had no effect. This observation is reminiscent of the often initially encouraging responses seen in patients, after which therapy-resistant regrowth is observed. It is important to mention that of the 24 mice, 16 had to be killed before the end of the experiment because they met the criteria set as humane end points, including ulceration and a tumor size >1.5 cm^3^.

To determine whether paclitaxel-loaded OPT-MSNs reduce side effects compared to free drug treatment, we used the collected blood samples to compare leukopenia and organ damage caused by systemic paclitaxel and OPT-MSN administration. Free-drug-induced systemic toxicity was evidenced by reduced white blood cell counts (Figure 4D), which were accompanied by increased lactate dehydrogenase (LDH; Figure 4E) and alanine aminotransferase (ALAT, a marker of hepatotoxicity; Figure 4F) levels. On the other hand, OPT-ADAM9-MSN-treated mice showed no systemic toxicity (Figure 4E). Importantly, hepatic accumulation of OPT-ADAM9-MSNs caused by repeated MSN injections (Figure 3B) did not result in hepatotoxicity, as evidence by plasma ALAT levels (Figure 4F). However, a reduced white blood cell count was observed after OPT-ADAM9-MSN treatment, although to a clearly lesser extent than that observed in free-drug-treated mice (Figure 4D). Taken together, these results show that OPT-MSNs reduce leukopenia and tissue damage compared to systemically administered free paclitaxel; however, no antitumor effect was observed upon OPT-MSN administration.

## 3. Discussion

Unfavorable cytotoxicity profiles limit the efficacy of current systemic treatments in PDAC patients. Improved strategies with more favorable toxicity profiles would reduce both treatment discontinuation events and dose reductions, ultimately leading to improved efficacy. Targeted delivery of chemotherapeutics to cancer cells specifically is an interesting approach to limit cytotoxicity to non-tumor tissue that may increase the number of patients that complete all planned cycles of chemotherapy, limiting PDAC mortality and morbidity with consequently reduced healthcare costs. We previously reported a novel ADAM9-responsive, protease-dependent drug delivery system for PDAC as a promising tool to reduce the cytotoxicity of systemic chemotherapy [9]. However, in this study, we show that our system was not as specific for PDAC cells as anticipated, based upon which we modified our linker for increased PDAC specificity. The optimized system efficiently induced PDAC cell death with limited bone marrow and neurotoxicity in vitro but failed to show antitumor activity in vivo.

Neuropathy and bone marrow toxicity are common side effects of paclitaxel treatment that hamper therapy efficacy in PDAC. Before pursuing in vivo experiments with our ADAM9-MSNs, we first assessed potential side effects using clinically relevant in vitro models for neuropathy and bone marrow toxicity. Rather surprisingly, based upon the presumed PDAC specificity of ADAM9 [9], paclitaxel-loaded MSNs administered at the IC20 for PDAC cells induced substantial death of SH-SY5Y and bone marrow cells. The lack of specificity is most likely not due to ADAM9 being expressed in neuronal or bone marrow cells but may actually be caused by the promiscuity of the linker. Several ADAM9-related proteases, such as MMP1, −2, −8, −9, −12, and −14 or ADAM8, −10, and −17, may also cleave this linker [15], and these proteases are not PDAC-specific (Figure 2A and [7,10]). The promiscuity of substrates assumed to be specific to one protease may not only pose problems with respect to the clinical translation of in vitro experiments for systems that employ protease substrates as gatekeepers for drug release purposes but may also hamper the clinical efficacy of protease-activated prodrugs [18] or protease-activated biologics [19] that rely on protease-specific drug activation.

To increase the specificity of the gatekeeper system, we next employed a different ADAM9-specific linker, which is supposedly not cleaved by metalloproteinases highly expressed in neuronal [20] and bone marrow cells [21] (i.e., MMP-2 and MMP-9). As expected, the optimized MSNs (OPT-MSNs) efficiently induced PDAC cell death with reduced cytotoxicity towards neuronal and bone marrow cells in vitro. Subsequent in vivo experiments employing syngeneic mice subcutaneously grafted with Kras^G12D^/p53^−/−^ (KP2) PDAC cells corroborated these results, as a decrease in leukopenia and organ damage was observed with OPT-MSN administration. As white blood cells express low ADAM9 levels and do not seem to cleave the ADAM9-responsive peptide linker (Figure 2A,H and [9]), leukopenia in the OPT-MSN-treated mice is most likely not due to off-target cytotoxicity by white-blood-cell-dependent cap removal and paclitaxel release. Leukopenia is probably caused by paclitaxel released by ADAM9-dependent cap removal in the tumor microenvironment, which is subsequently secreted into the bloodstream or by paclitaxel released by circulating tumor-derived ADAM9. Importantly and rather unexpectedly, we did not observe a significant antitumor effect despite the use of a treatment protocol that leads to efficient delivery of MSNs to the tumor. Although we do not have a proper explanation for the poor clinical translation, it is tempting to speculate that paclitaxel levels in the tumor did not reach the necessary threshold. This may be particularly likely, as MSN uptake by tumor cells was not optimal (i.e., around 50% of the tumor cells did not take up MSNs after repeated administration), and the fact that free paclitaxel treatment showed very limited therapeutic efficacy suggests that the therapeutic window of paclitaxel is rather narrow in our model. The incorporation of a tumor-targeting moiety into the OPT-MSNs via promising options such as TAB004 [22], V7-peptide [23], or cetuximab [24] could theoretically lead to increased MSN uptake in the tumor and consequent above-threshold paclitaxel levels to induce tumor cell death and subsequent tumor size reduction. However, the validity of this tantalizing hypothesis needs to be addressed in future studies.

The poor in vivo antitumor activity of the OPT-MSNs despite very efficient antitumor effects in vitro underscores the importance of considering adequate tumor physiology in preclinical experiments before pursuing clinical studies. In vitro experiments, although useful for proof of principle, do not capture the complex nature of tumor growth in vivo and lack key features such as stromal involvement and blood flow. Despite these obvious limitations of in vitro studies to predict clinical efficacy, most previously published papers on protease-responsive MSNs lack in vivo confirmation of antitumor activity despite suggesting their considerable potential for cancer therapy [25,26,27,28,29,30,31]. The disappointing data presented here emphasize the notion that although MSN-based tumor-targeting strategies may hold therapeutic potential, clinical translation does not seem as straightforward as anticipated [4,32,33]. Therefore, we propose that the field of cargo release by protease-responsive MSNs needs more extensive exploration of its therapeutic potential before we can expect it to reach clinical translation.

In conclusion, we generated highly specific ADAM9-responsive MSNs that very efficiently induced PDAC cell death with limited bone marrow and neurotoxicity in vitro. However, no antitumor activity was observed in vivo.

## 4. Materials and Methods

### 4.1. Cell Culture

Human PANC-1, Capan-1 (both ATCC, Manassas, VA, USA), SH-SY5Y (kindly provided by Dr. J. van Nes, Amsterdam UMC, Amsterdam, The Netherlands), and murine KP2 cells (derived from pancreatic adenocarcinomas from p48-Cre/LSL-Kras^G12D^/Tp53flox/flox mice kindly provided by Dr. DeNardo, Washington University Medical School, St. Louis, MO, USA) were grown in DMEM (Lonza, Basel, Switzerland) supplemented with 10% fetal calf serum, 100 units/mL penicillin, and 500 µg/mL streptomycin (all Lonza). SH-SY5Y culture medium was additionally supplemented with MEM non-essential amino acids (#11140050, ThermoFisher, Waltham, MA, USA). Cells were cultured in a humidified incubator at 37 °C under 5% CO_2_. All cell lines were tested for the absence of mycoplasma on a monthly basis.

### 4.2. Neurotoxicity Assay

A total of 1000 SH-SY5Y cells were seeded in a 96-well plate and, after 48 h, supplemented with all-trans retinoic acid (ATRA; #R2625-50MG, Merck, Rahway, NJ, USA) to a final concentration of 1 µM to induce differentiation into a more neuronal phenotype. After 48 h, various concentrations of capped paclitaxel-loaded ADAM9-MSNs or OPT-MSNs were administered. Three days later, 20 µL CellTiter-Blue (Promega, Leiden, The Netherlands) was added to the SH-SY5Y cells. After 3 h, fluorescence was measured at 590 nm on a Synergy HT plate reader.

### 4.3. Bone Marrow Cell Toxicity

Freshly excised C57BL/6 (Charles River Laboratories, Wilmington, MA, USA) murine hind limbs were washed in 70% ethanol for 30 s before transferring them to a Petri dish containing ice-cold sterile PBS. Bone marrow cells were collected in a 50 mL Falcon tube by cutting the femur and tibia at both ends with sterile scissors and flushing the bones with ice-cold sterile PBS using a 10 mL syringe containing a 25-gauge needle. After dispersing the bone marrow by sucking and expelling the cells with a 21-gauge needle, 5 volumes of erythrocyte lysis buffer (buffer EL) (#79217, Qiagen, Venlo, The Netherlands) were added to 1 volume of bone marrow cells. After 15 min incubation on ice, the cells were centrifuged at 400× *g* for 10 min at 4 °C, after which the supernatant was discarded. Next, two times the bone marrow volume of buffer EL was added to the cells, and they were briefly vortexed and centrifuged at 400× *g* for 10 min at 4 °C. After discarding the supernatant, the cells were resuspended in ice-cold sterile PBS and counted. A total of 90,000 cells were added to 3 mL mouse methylcellulose complete medium (#HSC007, R&D Systems, Minneapolis, MN, USA) containing varying concentrations of free paclitaxel, capped paclitaxel-loaded ADAM9-MSNs, or OPT-MSNs before plating 1 mL medium/well in a 12-well plate using a 16-gauge needle. After 7 days of incubation at 37 °C, the number of colonies was counted.

### 4.4. Assessment of Publicly Available Gene Expression Datasets

Datasets derived from the Gene Expression Omnibus (https://www.ncbi.nlm.nih.gov/gds, accessed on 28 December 2022) were analyzed using the R2 microarray analysis and visualization platform (http://r2.amc.nl, accessed between December 2022 and February 2023). Protease expression levels were obtained from cell line datasets (GSE36133, GSE57083, and E-MTAB-783) and an in vitro activated and differentiated human macrophage dataset (GSE46903).

### 4.5. Synthesis and Surface Modification of MSNs

MSNs were synthesized using the sol–gel emulsion described in [9]. In short, cetyltrimethylammonium bromide (CTAB, ≥99%, Sigma-Aldrich, Shanghai, China) served as a mesoporous template, followed by in situ polymerization using tetraethyl orthosilicate (TEOS, ≥99%, Sigma-Aldrich). After two hours, the surfactant CTAB was removed, and the resulting MSNs were filtered, washed, and dried under a high vacuum. Surface amine grafting was achieved through overnight refluxing by dried toluene (>99.3%, Honeywell Fluka, Seelze, Germany) and aminopropyl triethoxysilane (APTES, ≥99%, Sigma-Aldrich). Finally, the mixture was filtered and dried under vacuum to obtain MSN-NH_2_. Successful synthesis of the MNSs was assessed transmission electron microscopy as described in [9]. To protect the conformation of the peptide, interactions between the peptide and the MSNs were inhibited by modifying the MSNs with PEG_4_-N_3_. As described in [34], N_3_-PEG_4_-COOH (1 eq., >97%, Biomatrik, Jiaxing, Zhejiang, China), HATU (2 eq., 99%, Alfa Aesar, Kendel, Germany), and DIPEA, (3 eq., ≥99%, Sigma-Aldrich) were added to a suspension of MSN-NH_2_ in DMF (≥99.8%, Biosolve Chimie, Dieuze, France) and stirred overnight at room temperature, after which the MSNs were filtered and washed with water and ethanol to obtain MSN-PEG_4_-N_3_.

### 4.6. Synthesis, Conjugation, and Characterization of Linker Peptides

The synthesis and conjugation of linker peptides (shown in Table 2) was accomplished as described in [9]. First, amino acids were Fmoc-deprotected by 20% piperidine (Biosolve Chimie) in DMF, followed by amide coupling using protected amino acid, diisopropylcarbodiimide (DIC, 99%, Acros Organics, Budapest, Hungary), and Oxyma Pure (Carl Roth, Karlsruhe, Germany). All protected amino acids were acquired from Novabiochem (Darmstadt, Germany), except for Fmoc-propargyl-glycine and Fmoc-homophenylalanine, which were acquired from Glentham Life Sciences (Corsham, UK) and Carbolution (St. Ingbert, Germany), respectively. Fmoc-D-homophenylalanine was acquired from Alfa Aesar. Biotinylation of the peptide N terminus was achieved using biotin (3 eq., ≥99%, Sigma-Aldrich, lyophilized powder, St. Louis, MO, USA), HATU (3 eq., 99%, Alfa Aesar), and DIPEA (6 eq., ≥99%, Sigma-Aldrich). 

Next, the peptide linkers were cleaved from the resin with a mixture of trifluoroacetic acid (TFA, ≥99.5%, Biosolve Chimie), triisopropylsilane (TIPS, 98%, Sigma-Aldrich), and water (TFA:TIPS:H_2_O 95:2.5:2.5 *v*/*v*/*v*) and precipitated using cold diethyl ether (100%, VWR Chemicals BDH, Darmstadt, Germany). All peptides were purified using HPLC, and eluents consisted of 0.1% TFA in water (A) and 0.1% TFA in acetonitrile (B) for all peptides. LC-MS was used to assess the purity of the collected fractions. Pure fractions were pooled and lyophilized. 

Conjugation of MSNs with linker peptides was achieved using a copper-catalyzed click reaction. A reaction mixture consisting of anhydrous copper(II)sulfate (CuSO_4_, ≥98%, ThermoFisher), sodium L-ascorbate (≥99%, Sigma Aldrich), and tris(3-hydroxypropyltriazolylmethyl)amine (THPTA, Lumiprobe, Hunt Valley, MD, USA) in deionized water was flushed with nitrogen for 30 min, after which 10 µmol biotin-coupled linker peptides and 10 mg MSN-PEG_4_-N_3_ were added. The reaction mixture was then stirred overnight at room temperature in an inert atmosphere, after which the MSNs were centrifuged and washed with water and ethanol to obtain peptide-linker-functionalized MSNs.

Characterization of MSNs was performed as described in [9]. In short, Fourier transform infrared (FT-IR) spectroscopy and zeta potential (ZP) were used to confirm the successful synthesis and functionalization of the MSNs, and dynamic light-scattering (DLS) measurements were performed to assess the size of the MSNs. 

### 4.7. Paclitaxel Loading and Capping of MSNs

Peptide-linker-functionalized MSNs were loaded with paclitaxel by the adsorption equilibrium method [35] and capped with avidin as described in [9]. In summary, 1 mg of MSNs was added to 10 mg/mL paclitaxel (>99.5%, LC Laboratories, Woburn, ON, Canada) and incubated at 37 °C and 1200 rpm for 1.5 h. Loaded MSNs were centrifuged to measure the UV absorbance of the supernatant and determine the amount of loaded paclitaxel. Next, the paclitaxel-loaded MSNs were capped by adding 1.5 mg avidin (EMD Millipore, Burlington, MA, USA) for 30 min. After 30 min, the capped MSNs were washed twice with water and ethanol. Finally, the particles were redispersed in HBSS (Gibco, ThermoFisher) at a concentration of 1 mg/mL. Differences in loading efficiency were overcome by determining the IC50 or IC20 in PANC-1 cells for each batch of MSNs and applying these concentrations in comparable experiments.

### 4.8. PDAC Cytotoxicity Assays

PANC-1 (3500), Capan-2 (7500), and KP2 (3000) cells were seeded in a 96-well plate and left to attach overnight. After attaching, the cells were incubated with different concentrations of free paclitaxel, capped paclitaxel-loaded ADAM9-MSNs, OPT-MSNs, or uncleavable-OPT-MSNs for 72 h. After 72 h, PANC-1, Capan-2, and KP2 cells were washed and incubated with crystal violet (0.5% crystal violet, 3% formaldehyde in dH_2_O) at room temperature. To solubilize the formed crystals, the crystal violet solution was aspirated after 20 min, cells were washed thrice with tap water, and 100 µL DMSO (Merck) per well was added. After 20 min incubation on a plate shaker at room temperature, the absorbance was measured at 600 nm on a Synergy HT plate reader. 

### 4.9. Synthesis of ATTO488-Labeled MSNs for In Vivo Biodistribution Experiments

MSNs containing thiol groups in the particle core and amine groups on the particle surface were synthesized using the procedure described in [25]. A mixture of 0.54 g (2.61 mmol) tetraethyl orthosilicate (TEOS, ≥99%, Sigma-Aldrich), 37.3 mg (0.16 mmol) (3-Mercaptopropyl)triethoxysilane (MPTES, ≥80%, Sigma-Aldrich), and 4.77 g (31.87 mmol) triethanolamine (TEA, >99.0%, Sigma-Aldrich) was heated under static conditions at 90 °C for 20 min in a Teflon-lined stainless-steel hydrothermal vessel. A solution of 0.89 mL (0.61 mmol) cetyltrimethylammonium chloride (CTAC, 25 wt% in H_2_O, Sigma-Aldrich) and 33.3 mg (0.9 mmol) ammonium fluoride (NH_4_F, 100%, Boom B.V., Meppel, The Netherlands) in 7.2 mL (0.4 mmol) H_2_O was preheated to 60 °C and added to the TEOS solution. This reaction mixture was stirred for 20 min at 700 rpm while cooling to room temperature. Next, 49 µL (0.31 mmol) TEOS was added in four equal increments every 3 min. After 30 min, 6.9 µL (30.8 µmol) TEOS and 7.2 µL (30.8 µmol) aminopropyl triethoxysilane (APTES, ≥99%, Sigma-Aldrich) were added to the reaction mixture, which was then allowed to stir at room temperature overnight. The MSNs were collected by centrifugation, washed with water and ethanol, and dried on a Buchner funnel under a high vacuum. The surfactant CTAC was removed by refluxing the MSNs for 45 min in 33 mL ethanol containing 0.67 g (8.4 mmol) ammonium nitrate (NH_4_NO_3_, ≥98%, Honeywell Fluka). After collecting the MSNs by centrifugation, they were washed with absolute ethanol and refluxed in 3.33 mL hydrochloric acid (37%, Fluka) and 30 mL ethanol. The resulting SH-MSN-NH_2_ was then collected by centrifugation, washed with ethanol, and dried under nitrogen flow.

SH-MSN-NH_2_ was labeled with ATTO 488 (Atto-tec, Siegen, Germany) by a thiol-maleimide reaction [36]. In short, 30 µL of an ATTO 488 maleimide solution (5 µg/µL anhydrous dimethylformamide (DMF, ≥99.8%, Biosolve Chimie)) was added to a suspension of 30 mg SH-MSN-NH_2_ in 30 mL absolute ethanol and stirred overnight. These ATTO-488-labeled MSNs (ATTO-MSNs) were then collected by centrifugation, washed several times with absolute ethanol, dried, and used for biodistribution experiments. ATTO-488-MSNs were characterized using similar techniques as described above.

### 4.10. Animals

Female C57BL/6 mice (Charles River Laboratories, Wilmington, MA, USA) were housed at the Amsterdam University Medical Center’s (AUMC) animal facility. All animals had access to water and food ad libitum. All animal experiments were approved by the Institutional Animal Care and Use Committee of the AUMC according to protocol DIX-19-9064.

### 4.11. In Vivo Biodistribution of MSNs

Subcutaneous xenograft tumors were generated in the left flank of female C57BL/6 mice by injecting 1 × 10^6^ KP2 cells in a mixture of DMEM (Lonza) and Matrigel (Corning, Corning, NY, USA) in a 1:1 ratio with a total volume of 100 µL. Tumor growth was measured twice a week using a caliper. To determine the optimal dose of MSNs to administer, 12.5 mg/kg, 25 mg/kg, or 50 mg/kg ATTO-labeled MSNs were intravenously injected by tail injection three weeks after tumor injection. Four hours later, the mice were killed, and the lungs, liver, kidney, colon, hind limbs, and tumor were removed. Tumor cell suspensions were prepared mechanically without enzymatic treatment, and ATTO-MSN fluorescence was measured in technical triplicates at 488 nm using a CytoFLEX S (Beckman Coulter, Brea, CA, USA). Next, we set out to determine the optimal treatment schedule. First, subcutaneous tumors were generated as previously mentioned. After two weeks, 50 mg/kg ATTO-labeled MSNs were intravenously injected, and mice were killed after 24 and 48 h. One treatment group received a second injection after 48 h and were killed 24 h after the second injection. All major organs were removed, dissociated, and measured as previously described. All generated data were analyzed using FlowJo Software (BD Biosciences, San Jose, CA, USA). The gating strategy used for analysis is shown in Appendix A.

### 4.12. Antitumor Efficacy of Paclitaxel-Loaded OPT-MSNs

First, we loaded 60 separate batches of 1 mg, which increased loading efficiency compared to larger batches. To confirm the successful loading of paclitaxel before pooling the loaded batches, several randomly selected batches of OPT-MSNs were administered to PANC-1 cells in vitro. As all loaded batches resulted in approximately 80% cell death (Appendix A), indicating the successful and uniform loading of paclitaxel, we pooled the samples and assessed the cytotoxicity in vivo. Next, 25000 KP2 cells in a 1:1 mixture of DMEM and Matrigel were subcutaneously injected in the left flank of C57BL/6 mice. One week after subcutaneous injection, the mice were randomly assigned to receive intravenous injection of PBS, 20 mg/kg free paclitaxel, or 50 mg/kg paclitaxel-loaded OPT-MSNs in a total volume of 100 µL. All mice were treated twice weekly for 4 weeks, with a total of 7 injections. Body weight and tumor growth were measured three times a week using a scale and caliper. Mice were killed two days after receiving the last injection or upon reaching humane end points specified as ulceration, tumor size >1.5 cm^3^, acute weight loss >15% within two days, chronic weight loss of >15% combined with visual discomfort, or chronic weight loss of >20% without any signs of discomfort. Blood collected by heart puncture was centrifuged at 3500 rpm for 10 min at 4 °C, and plasma was collected and stored at −70 °C. To assess leukopenia, the blood cell pellet was incubated twice with Buffer EL for 15 min on ice and centrifuged at 400× *g* for 10 min at 4 °C. The remaining white blood cells were resuspended in 10 mL ice-cold sterile PBS and counted.

### 4.13. Biochemical Analysis

Plasma creatinine, alanine aminotransferase (ALAT), and lactate dehydrogenase (LDH) levels were measured at the clinical diagnostics laboratory of the AUMC.

## Figures and Tables

**Figure 1 ijms-24-10704-f001:**
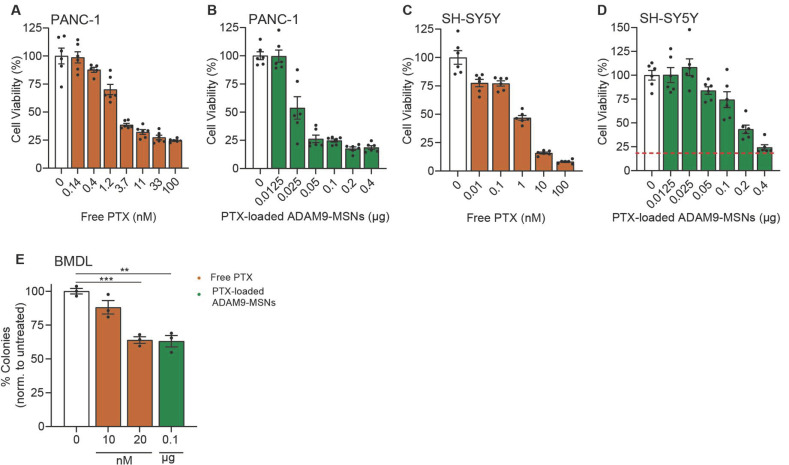
Paclitaxel-loaded ADAM9-MSNs evoke neurotoxicity and bone marrow toxicity in preclinical models in vitro. IC50 curve in PANC-1 (**A**) and SH-SY5Y (**C**) cells after 72 h of free paclitaxel administration (data modified from a previous publication [9]). Cytotoxicity of paclitaxel-loaded ADAM9-MSNs in PANC-1 (**B**) and SH-SY5Y (**D**) cells after 72 h. Data are shown as the mean of one representative experiment with *n* = 6. Results are normalized to untreated controls. Microscopic images resembling diverse bone-marrow-derived colonies from representative colony-forming assays (**E**). Cytotoxicity of free paclitaxel and paclitaxel-loaded ADAM9-MSNs on bone marrow cells from colony-forming assays after 7 days. Data are shown as the mean of one representative experiment with *n* = 3. Levels of significance: ** *p* < 0.01 and *** *p* < 0.001.

**Figure 2 ijms-24-10704-f002:**
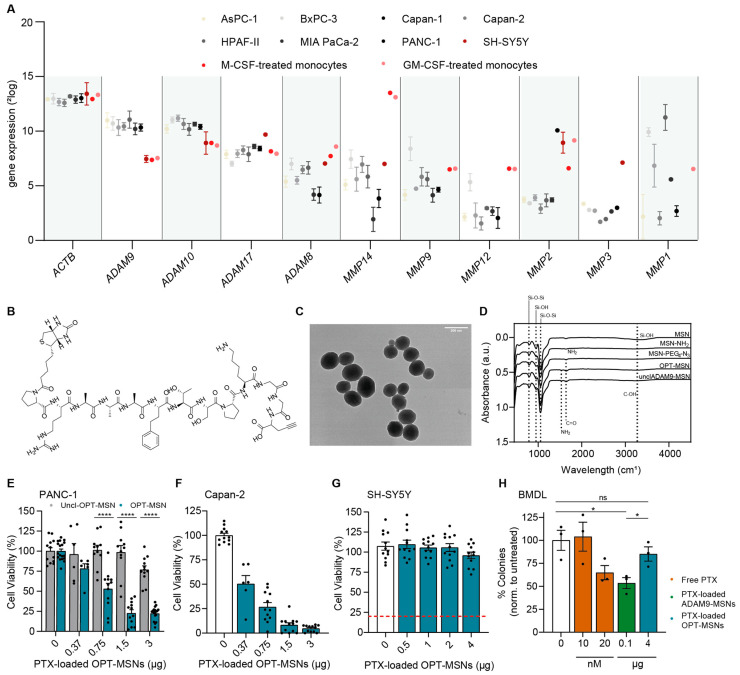
MSNs with a modified ADAM9 linker have increased PDAC specificity and reduced general toxicity. (**A**) Expression levels of proteases capable of cleaving the ADAM9 linker. Data were extracted from publicly available gene expression datasets (GSE36133, GSE57083, GSE46903, and E-MTAB-783). (**B**) Molecular structure of modified ADAM9 linker (OPT linker). (**C**) Transmission electron microscopic images of peptide–biotin-coupled MSNs. (**D**) Fourier transform infrared spectra of MSNs, several modified intermediates, and final products OPT and uncleavable OPT-MSNs. (**E**) Cytotoxicity of paclitaxel-loaded OPT and uncleavable OPT-MSNs in PANC-1 cells after 72 h (*n* = 12). (**F**) Cytotoxicity of paclitaxel-loaded OPT-MSNs in Capan-2 cells (*n* = 12) and SH-SY5Y cells ((**G**), *n* = 12) after 72 h and in bone marrow cells ((**H**), *n* = 3) after 7 days. Data are normalized to untreated controls. Levels of significance: ns = not significant, * *p* < 0.05 and **** *p* < 0.0001.

**Figure 3 ijms-24-10704-f003:**
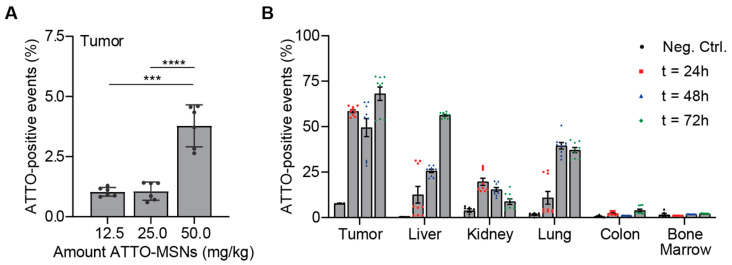
Intravenous injection of fluorescent MSNs results in significant tumor accumulation in vivo. (**A**) Determination of optimal dose administration in vivo. (**B**) In vivo biodistribution of fluorescent MSNs after repeated administration. Measurements were performed in technical replicates. Levels of significance: *** *p* < 0.001 and **** *p* < 0.0001.

**Figure 4 ijms-24-10704-f004:**
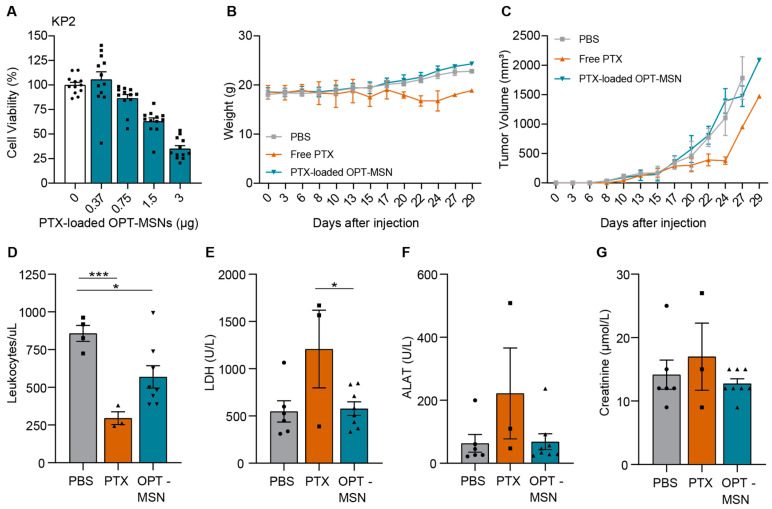
Paclitaxel-loaded OPT-MSNs reduce neutropenia and organ damage but show no antitumor effect in vivo. (**A**) Toxicity of paclitaxel-loaded OPT-MSNs in KP2 cells 72 h after administration (*n* = 12). (**B**) Weight of mice during the experiment. (**C**) Tumor volume (mm^3^) plotted over time. Data analysis was complicated by the fast-growing nature and ulcerative properties of KP2 cells, causing the sacrificing of 16 mice before the end of the experiment because they met the criteria set as humane end points. Both ulceration and reaching a tumor size of >1.5 cm^3^ led to the killing of eight mice, and two mice reached both humane end points simultaneously. (**D**) Blood leukocyte counts at sacrifice. Plasma LDH (**E**), ALAT (**F**), and creatinine (**G**) levels as measured by HPLC following standardized clinical guidelines of the AUMC. Levels of significance: * *p* < 0.05 and *** *p* < 0.001.

**Table 1 ijms-24-10704-t001:** Hydrodynamic size, PDI, and zeta potential of MSNs, MSN-NH2, MSN-PEG4-N3, peptide-linker-functionalized MSNs, and ATTO488-MSNs in water measured by dynamic light scattering.

Sample	Size ± St.Dev. (d.nm)	PDI	ZP ± St.Dev. (mV)
MSNs	223.6 ± 62.11	0.36	−18.93 ± 0.12
MSN-NH_2_	191.5 ± 42.20	0.69	19.13 ± 0.73
MSN-PEG_4_-N_3_	233 ± 59.38	0.38	0.56 ± 4.26
ADAM9-MSN	189	X	22.1 ± 0.8 [9]
OPT-MSN	210.6 ± 60.78	0.14	30.53 ± 1.24
UNCL-ADAM9-MSN	238.1 ± 91.56	0.51	29.27 ± 1.096
ATTO488-MSN	185.0 ± 3.52	0.10	29.9 ± 0.15

**Table 2 ijms-24-10704-t002:** Overview of peptides and characteristics.

Peptide	Sequence	Theoretical Mass (Da)	Measured *m*/*z*	Purity
ADAM9-biotin	Biotin-SPLAQAVRSSK	1577.80	1576.66	≥98%
OPT- biotin	Biotin-PRAAAF*TSPKGGG*	1493.74	1492.43	≥98%
UNCL-ADAM9-biotin	Biotin-PRAAA*F^¥^TS^¥^PKGGG*	1493.74	1492.64	≥99%

F* = homophenylalanine; G* = propargylglycine; A* = D-alanine; F^¥^ = D-homophenylalanine; S^¥^ = D-serine.

## Data Availability

The data presented in this study are available from the corresponding author upon reasonable request.

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
