# Peer review of "Preclinical Assessment of ADAM9-Responsive Mesoporous Silica Nanoparticles for the Treatment of Pancreatic Cancer"

_ijms, 2023, doi:10.3390/ijms241310704_

Round 1

Reviewer 1 Report

This study evaluated the efficacy of AMAM9-MSN which is a PDAC-specific drug delivery system. Such studies are important because pancreatic cancer has a lot of stroma and delivery of drugs to the tumor is a challenge.

Unfortunately, PTX-loaded MSN did not show to reduce the tumor volume (in vivo), MSN had confirmed a certain effect in vitro. It may be possible to expect an effect by changing to a drug other than PTX. The methods and results are well described, and there seems to be no problem with the publication of the current version.

This study evaluated the efficacy of AMAM9-MSN which is a PDAC-specific drug delivery system. Such studies are important because pancreatic cancer has a lot of stroma and delivery of drugs to the tumor is a challenge.

Unfortunately, PTX-loaded MSN did not show to reduce the tumor volume (in vivo), MSN had confirmed a certain effect in vitro. It may be possible to expect an effect by changing to a drug other than PTX. The methods and results are well described, and there seems to be no problem with the publication of the current version.

Author Response

Dear reviewer,

Please find attached our author's reply to the review report.

Thanks in advance,

Etienne Slapak

Reviewer 2 Report

Etienne et al. reported that “Evaluation of ADAM9-Responsive Mesoporous Silica Nanoparticles for the Treatment of Pancreatic Cancer. The authors evaluated the Pancreatic adenocarcinoma (PDAC) -specific drug delivery system using mesoporous silica nanoparticle (MSN) with a protease linker that is specifically cleaved by PDAC cells. It’s interesting to make mesoporous silica nanoparticles and attached them to the drug and evaluated the targeting therapy. The experimental design is good, and the execution of the experiments and their results are good. Here are a few comments to the authors to improve the quality of the manuscript.

Comments

1.     In Figure 1, Why the authors missed the control group or without PTX in A and C figures

2.     What is ADAM9, please expand it.

Author Response

(The authors gave the same response as above.)

Reviewer 3 Report

In this manuscript, the authors prepared ADAM9-responsive nanoparticle loading PTX. The nanoparticle suppressed the side effect, but unfortunately the efficacy was not enough in tumor-bearing mice. Some parts of this study is preliminary and some aspect should be addressed before publication.

Major points

1. In introduction section, the information on ADAM9-responsive peptide is lacking. Peptide sequence and peptide density on MSN should be descried.

2. Did the author check the toxicity of empty MSN? Is it possible that cytotoxic effect of MSN contributes to enhanced cytotoxicity of PTX-MSN?

3. In figure 2, SH-SY5Y cells express protease that cleave ADAM9 123 linker, which contribute to cytotoxicity. However, did the author check the difference of cytotoxicity of PTX-MSN without peptide linker in SH-SY5Y cells and PDAC cell lines? I wonder whether sensitivity to paclitaxel or the difference of uptake ability of MSN might affect the cytotoxicity between SH-SY5Y cells and PDAC cell lines.

4. In figure 3, the author mentioned repeated injection increased the tumor accumulation, but also increased hepatic accumulation, which might cause hepatotoxicity. In addition, even after second injection, half tumor cells did not take up the particles, which leads insufficient antitumor effect. It is better to discuss about this issues.

5. In figure 4, in vitro cytotoxic effect of free PTX against KP2 should be presented.

6. Unfortunately, PTX-MSN did not show antitumor effect in SK2-bearing mice although free PTX did. Further studies are need to clarify why PTX-MSN is not effective. For example, did the authors check the accumulation of PTX itself in tumor? Also did the authors check the expression of ADMA9 in vivo?

Minor points

1. Did the author prepare tumor cell suspension without enzymatical treatment?

2. In figure 3, y-axis should be ATTO-positive cells, but not FITC-positive cells.

Author Response

(The authors gave the same response as above.)

Round 2

Reviewer 3 Report

I have no additional comments.